# Microbiota and Pain: Save Your Gut Feeling

**DOI:** 10.3390/cells11060971

**Published:** 2022-03-11

**Authors:** Chiara Morreale, Ilia Bresesti, Annalisa Bosi, Andreina Baj, Cristina Giaroni, Massimo Agosti, Silvia Salvatore

**Affiliations:** 1Department of Woman and Child, ASST-Settelaghi, University of Insubria, 21100 Varese, Italy; cmorreale@studenti.uninsubria.it (C.M.); massimo.agosti@uninsubria.it (M.A.); silvia.salvatore@uninsubria.it (S.S.); 2Department of Medicine and Surgery, University of Insubria, 21100 Varese, Italy; a.bosi@uninsubria.it (A.B.); cristina.giaroni@uninsubria.it (C.G.)

**Keywords:** pain, microbiota, probiotics, gut-brain axis

## Abstract

Recently, a growing body of evidence has emerged regarding the interplay between microbiota and the nervous system. This relationship has been associated with several pathological conditions and also with the onset and regulation of pain. Dysregulation of the axis leads to a huge variety of diseases such as visceral hypersensitivity, stress-induced hyperalgesia, allodynia, inflammatory pain and functional disorders. In pain management, probiotics have shown promising results. This narrative review describes the peripheral and central mechanisms underlying pain processing and regulation, highlighting the role of the gut-brain axis in the modulation of pain. We summarized the main findings in regard to the stress impact on microbiota’s composition and its influence on pain perception. We also focused on the relationship between gut microbiota and both visceral and inflammatory pain and we provided a summary of the main evidence regarding the mechanistic effects and probiotics use.

## 1. Introduction

Pain is a natural protective mechanism of the body that arises from nociceptors and involves the interaction of different neuroanatomic and neurochemical systems. According to the International Association for the Study of Pain (IASP): “Pain is an unpleasant sensory and emotional experience associated with actual or potential tissue damage, or described in terms of such damage” [1]. Experience of pain is the result of the interplay between several compartments: receptors, neurotransmitters involved in the regulation of pain perception, pain-related emotions and memory [2]. Acute pain is an alarm system that protects us from tissue damage. Chronic pain (which has a duration of more than three months) is a persistent pain associated with injury, disorders or diseases (i.e., arthritis, functional gastrointestinal disorders, inflammatory bowel diseases, diabetes or tumor growth). This can be a consequence of damage to nerve fibers which leads to an increased spontaneous firing or alterations in their conduction or neurotransmitter properties [3]. Chronic pain is considered to be a disease state and the Institute of Medicine claims that more than 100 million Americans suffer from chronic pain [4]. Recently, a growing body of evidence has shown that the gut microbiota may play a role in modulating visceral pain and inflammatory and neuropathic pathways [5]. Although still limited, emerging research reports the involvement of the gut microbiota in the release of signal molecules (i.e., metabolites, neurotransmitters, neuromodulators), which are directly involved in pain transmission and modulation [5]. In this narrative review, we aim to provide a comprehensive overview regarding the influence of gut microbiota on pain perception and the potential application of probiotics in pain management.

## 2. The Pathway of Pain at a Glance

The pain pathways are involved in four main physiological steps: transduction, transmission, modulation and perception. Through this complex network, pain receptors, called nociceptors (from the Latin word “noxa” that means damage and receptors), send the hurt signal to the spinal cord and then to the central nervous system (CNS). At this level, the stimulus is fully processed, determining the perception of intensity of pain and a related efferent response and behavioral reactions [1,2,4,6]. Pathologic painful conditions may also be secondary to disruption of the peripheral and central nociceptive process or secondary to psychological conditions [4].

Pain starts from activation of nociceptors, which are the sensors of potential or real damage located in the skin, joint capsule, periosteum, ligaments, muscles, cornea, dental pulp and internal organs. Nociceptors are free nerve endings that could be stimulated by biological, electrical, thermal, mechanical and chemical (such as histamine, substance P, bradykinin, acetylcholine and prostaglandins) stimuli [2]. The noxious stimulus is then transduced into an electrical signal generating an action potential along primary afferent neurons.

From a physiological point of view, pain may be experienced when nociceptor stimulation is intense enough to activate myelinated A (β or δ) or unmyelinated C fibers [1,6]. Several target receptors and ion channels located on peripheral terminals and along axons of primary afferent neurons are involved in pain stimulus transduction. The main nociceptors are:TRP (transient receptor potential) family which includes different thermosensitive ion channels (TRPV1, TRPV2, TRPV3, TRPV4, TRPM8 and TRPA1) that are activated by distinct thermal thresholds [3,7]. TRPV1 is also a vanilloid receptor for capsaicin and protons (acid content), whilst TRPM8 and TRPA1 are cold- and menthol-sensitive channels [3].Two members of KCNK potassium channel family (KCNK2, also called TREK-1 and KCNK4 or TRAAK), which are expressed in a subset of C-fiber nociceptors. These receptors can be modulated by pressure, temperature and pharmacological stimuli and, at the same time, they can modulate nociceptor excitability [3,8].Acid-sensing ion channels (ASICs) which mediate acid or chemical stimulus [4].

Voltage-gated ion channels (i.e., voltage-gated sodium: Nav 1.7, Nav 1.8, potassium or voltage-dependent calcium channels) convey nociceptor signals to electrical signals to be propagated through the primary afferent neuron to the synapses located in the dorsal horn of the spinal cord. Interestingly, nociceptors may also be activated by inflammatory mediators that are secreted at the site of injury, such as bradykinin, prostaglandins that enhance nociceptor sensitivity by blocking potassium channels, serotonin, histamine and nerve growth factor, which modulate nociceptor activation via TrKA receptors. Mediators composing this inflammation are able either to excite nociceptors or to lower their activation threshold. These molecules may also be involved in the development of neurogenic inflammation, where active nociceptors release neurotransmitters (i.e., substance P, SP, calcitonin gene-related peptide, CGRP, vasoactive intestinal peptide, VIP) from peripheral terminals. The release of these peptidergic neurotransmitters may be associated with vasodilation inducing extracellular leakage of fluid and proteins. Overall, these inflammatory conditions result in immune system activation that in a vicious circle enhances inflammation and ultimately leads to pain perception [1,3,4].

Two types of afferent fibers are able to transmit the impulse to the dorsal horn: medium diameter myelinated fibers (A-δ) that mediate well-localized or fast pain; and small diameter unmyelinated C fibers, that mediate poorly localized pain, also defined as secondary pain (dull burning or aching sensations). Aβ fibers are myelinated and transmit from the stretch receptor. These first-order afferent fibers represent the ramification of pseudo-unipolar neurons, whose soma is located in the somatic or visceral ganglia and bring the impulse from the peripheral terminals to the dorsal horn.

The fast transmission through myelinated Aẟ fibers generates a quick response of the body (“escape” or “fight”). Conversely, C fibers are slow conducting, are susceptible to damage and some of them are organized into “nets”. Nociceptive C-fiber terminals express, besides the multiple types of ion channels described above, opioid receptors that could be activated by endogenous (i.e., endorphins) and exogenous opioids. Endorphins are important for pain relief and their secretion is governed by the descending modulatory pain system which influences nociceptive input to the spinal cord [1,2].

In the dorsal horn, primary afferent neurons release neurotransmitters (i.e., glutamate, VIP, somatostatin, CGRP and SP) that activate the secondary somatosensory neuron. In the dorsal horn of the spinal cord, anatomically and electrophysiologically different laminae receive afferent fiber projections:○Aẟ fibers project to lamina I and deeper lamina V;○Aβ fibers end at laminae III and IV;○C fibers project to laminae I and II. 

Electrophysiological analyses demonstrate that spinal cord neurons of lamina I are responsive to noxious and thermal stimuli which correspond to the nucleus posteromarginalis. These axons join the controlateral spinothalamic tract. Neurons of lamina V receive both non-noxious and noxious input through direct (monosynaptic) Aẟ afferent fibers and indirect (polysynaptic) C afferent fibers. For this reason, neurons of lamina V are called wide dynamic range neurons. They receive information from visceral organs and are able to respond to a different range of stimulus intensities. Lamina II corresponds to the substantia gelatinosa and is important in the modulation of sensory pain input, having high levels of SP and opioid receptors. This structure receives and integrates inputs from peripheral sensory fibers (Aẟ, Aβ and C) to abrogate or transmit information through the second neuron of the ascending anterolateral pathways to the brainstem and brain. Moreover, the same structure receives descending pathways from the CNS for efferent outputs.

Gracile and cuneate fasciculi ascend through the dorsal column of the spinal cord. They carry information on two-point discrimination, fine touch sensation, pressure, vibration and proprioception. The lateral spinothalamic pathway (or neospinothalamic tract) carries information about pain, temperature and crude touch information from somatic and visceral structures. It is composed of crossed second-order axons and it terminates in the reticular formation of the brainstem, in the midbrain periaqueductal gray or in the posterolateral nucleus or intralaminar nucleus of the thalamus. The tertiary axons from the thalamus project to the posterior limb of the internal capsule, ending in the postcentral gyrus and posterior paracentral lobule of the parietal cortex. This cortical area is organized in a somatotopic map for an accurate localization of pain and receives information about sharp pain. Axons from the intralaminar nuclei end in the insula and rostral cingulate gyrus. These areas receive information about dull or deep pain and they are responsible for poor localized pain which is associated with emotions. Anterior spinothalamic tract (or paleospinothalamic tract) carries mostly pain, temperature and crude touch information in an ascending fashion to the brainstem and diencephalon. This tract includes crossed and uncrossed fibers which make synapses in the periaqueductal gray, reticular formation and tegmentum, ultimately projecting to the thalamus, secondary somatosensory cortex, cingulate and insula. 

The descending nociceptive pathways, through which the brain actively regulates the sensory transmission from the medulla, originate in the rostroventral medulla, in some brainstem nuclei (nucleus tractus solitarius, parabrachial nucleus), in the dorsal reticular nucleus, hypothalamus and cortex. They interact with afferent axons, interneurons and projection neurons of the spinal cord. Though this crosstalk, these pathways are able to suppress or enhance passage of nociceptive information to some structures involved in secondary processing (i.e., periaqueductal gray, thalamus, hypothalamus, parabrachial nuclei, nucleus tractus solitarius, amygdala, etc.) [1,4,9].

### Pain Processing

Pain perception depends on psychological processes and emotional, physiological and behavioral reactions. For this reason, pain is usually considered a multimodal condition [1,9].

The individual’s attention to pain plays a central role in pain perception. Therefore, attentional modulation of pain experience may induce changes in the pain neuromatrix activation. It is well known that lack of attention reduces pain-related activation in somatosensory cortices, insula, thalamus and other brain regions [1]. Furthermore, inattention triggers a strong activation of the prefrontal cortex, anterior cingulate cortex and periaqueductal gray, suggesting an interaction between brain regions involved in attentional modulation of pain and the descending pain pathways. In contrast, hypervigilance for pain amplifies pain intensity [1].

Cognitive appraisal of pain allows conscious or unconscious evaluation of sensory signals. The ability to interpret a body sensation as a threat partly depends on whether the subject believes themselves to be able to deal with this sensation. Therefore, pain intensity is reduced when pain is perceived as controllable. Ventrolateral prefrontal cortex activation is associated with a more controllable pain and with minor subjective pain intensity [1,10].

Nonetheless, emotional reactions to pain can induce some autonomic, endocrine and immune responses which could amplify the pain sensation through psychophysiological pathways. Pain inputs from viscera and muscles may stimulate cardiac vagal premotor neurons, causing hypotension, bradycardia and hyporeactivity. Additionally, proinflammatory cytokines and cortisol are released during a negative emotion and may enhance nociception [1,11].

It is noteworthy that the memory of pain may increase pain perception. Early life repeated pain experiences may lead to anticipatory pain behaviors and hyperalgesia [12]. Nevertheless, individual coping and behavioral reactions to pain may alleviate, exacerbate or prolong pain experience.

## 3. Microbiota and Gut-Brain Axis

The gut-brain axis is a bidirectional communication system between the gastrointestinal tract (GIT) and central nervous system (CNS) which integrates neural, hormonal and immunological signals (Figure 1).

Legend: NTS, nucleus of the solitary tract; NVG, nodose vagal ganglion; FFAR, free fatty acid receptor; DRG, dorsal root ganglion; MP, myenteric plexus, IPAN, intrinsic primary afferent neurons, SMP, submucosal plexus, EEC, enteroendocrine cell; EC, enterochromaffin cells, SCFA, short chain fatty acid [13].

A fundamental peripheral neuronal component is represented by the enteric nervous system (ENS), also known as “second brain”, which consists of intrinsic enteric neurons and glia innervating different lamina propria of the mucosa and the inner muscularis propria. This intrinsic neuronal network allows the GI tract to partially maintain its function in the absence of inputs from the CNS [14]. The human ENS is formed of about 200–600 million neurons, the same number composing the spinal cord. Four major neuron types have been classified according to the neurochemical coding, morphology and function: intrinsic primary afferent neurons (IPANs), interneurons, excitatory and inhibitory motor neurons contributing to the formation of the subserous, myenteric and submucosal plexuses [15]. IPANs detect diverse stimuli (i.e., chemical, including microbial metabolites, and mechanical) within the mucosa and muscularis propria and initiate motor, secretory and vasomotor local reflexes, as well as longer reflexes involving the CNS [16]. IPAN nerve endings do not extend into the gut lumen because they are separated by a continuous lining of epithelial cells. Enteric sensory neurons indirectly receive the information through hormones and neurotransmitter released by enteroendocrine (EEC) and enterochromaffin (EC) cells. Hormones such as cholecystokinin (CCK) are implicated in visceral pain; glucagone-like peptide 1 (GLP-1) regulates appetite and feeding behaviors, peptide YY (PYY) and serotonin (5HT). One of the best described examples is represented by the paracrine secretion of 5-HT from EC cells, which, by activating its own receptors on intrinsic and extrinsic primary afferent nerve terminals laying in proximity of ECs, triggers enteric reflexes mediating peristalsis, secretion, inflammation and pain perception [17]. Furthermore, 5-HT, via 5-HT4 receptor activation, has been shown to play a key role in ENS maturation [18]. Changes in 5-HT secretion by ECs have no direct consequence on the CNS, since the biogenic amine does not cross the blood-brain barrier (BBB) [19]. However, 5-HT released from ECs can potentially participate in brain-gut communication by modulating extrinsic afferent activity to the nucleus of the solitary tract and the dorsal raphe nucleus involved in the modulation of emotion-regulating brain networks that influence mood. In the CNS, 5HT is involved in the modulation of mood, behavior and cognitive functions and disruption of 5-HT homeostasis along the microbiota-gut-brain axis may participate in both dysmotility and development of mood disorders [17]. Afferent and efferent neurons, within the parasympathetic (vagal) and sympathetic (splanchnic and pelvic spinal pathways) branches of the autonomic nervous system (ANS) represent the main neuronal conduit participating in the two-way communication system between the gut and the brain [20]. The vagal extrinsic pathway allows the CNS to control the GIT, providing direct and indirect changes in the activity of intrinsic nervous circuits [21,22]. Vagal terminals impinge on enteric ganglia, smooth muscle cells and in the mucosal layers, and their activation induces an inhibitory modulation of chemical or mechanical insults [23]. Interestingly, as part of a reciprocal homeostatic relationship, the ANS can activate ECCs to release 5-HT in the gut lumen, which may then influence different cellular players of the enteric microenvironment, including the gut microbiota [22,24]. 

The gut microbiota is made of bacterial cells belonging to approximately 2000 species, and other microorganisms, such as viruses, archaea, fungi and protozoa, represent the most abundant microbial population in the human body [25,26]. Recent evidence has shown its relevant implication in gut-brain communication. The microbiota-gut-brain axis is essential for maintaining gut homeostasis by controlling some physiological functions of the GIT (i.e., secretion, motility, sensory, autonomic and secretory functions) [27]. Gut microbiota is a dynamic entity: its complexity and diversity are established in the first few years of life and are dependent on external factors such as vaginal delivery or caesarean section, diet, stress, infections and antibiotic medication [14,28,29]. It stabilizes after the first three years of life and maintains its characteristics during the lifespan unless different host factors (i.e., diet, disease, drugs, inflammation) occur. These alterations can lead to detrimental changes in the quantitative and qualitative composition of microbiota, defined as dysbiosis [27,30]. The gut microbiota, via its metabolites, is able to communicate with the CNS, through neural (i.e., vagus, ENS and spinal nerves), endocrine (cortisol) and immune (cytokines) pathways. In this latter regard, microbial inputs are considered fundamental for the development and function of the peripheral immune system and for the maturation and activation of microglia (innate immune cells of the brain). It is noteworthy that alteration in microglia function has been linked to stress and behavioral and neurodegenerative disorders (i.e., depression, anxiety and autism spectrum disorder). Therefore, cytokines and chemokines can be produced by the brain’s resident immune cells or arrive at the brain through direct transport across the blood-brain barrier (BBB). The permeability of BBB is influenced by the gut microbiota and inflammation; infections or autoimmune disease can modify the BBB integrity, causing a major translocation of microbial products to the brain. This is linked to the development of some neuropathological conditions, suggesting the potential impact of connections between systemic immunity and brain outcomes [30]. 

Innate immune receptors (Toll-like receptors (TLRs)) are important for sensing components of microbial cells, such as lipopeptides, peptidoglycans, glycolipids and lipopolysaccharides (LPS), and are also defined as microbial-associated molecular patterns (MAMP) receptors. Until now, thirteen TLRs have been identified in mammals and are expressed by a variety of cells, including ECs and ECCs, immunocytes, neurons and glial cells, both in the peripheral and central nervous system [14]. In the ENS, TLR2 and TLR4 expressed on both neurons and glial cells have fundamental roles in the regulation of the gastrointestinal functions [31,32]. Germ-free mice, antibiotic-treated mice and TLR2-/- and TLR4-/- mice showed a similar pattern of small intestine and colonic motor changes with alterations of myenteric neuron chemical coding and gliosis [32,33,34]. Recent evidence suggests that in the bidirectional communication between the peripheral neuronal immune and endocrine systems and the CNS, TLRs, namely TLR4, may be involved in the development of emotional regulation and stress-induced adaptations [35].

Microbes are able to locally synthesize neurotransmitters (GABA, noradrenaline, serotonin and dopamine, glutamate) and metabolites, such as short chain fatty acids (SCFAs) and tryptophan metabolites, such as 5-HT, kynurenines, tryptamine and indolic compounds, that are involved in microbiota-gut-brain communication [14,36,37] (Figure 1). SCFAs include butyrate, propionate, acetate and valerate which are produced by microbial fermentation of dietary polysaccharides in the cecum and colon [23]. Although conflicting evidence is emerging on the role of SCFAs in visceral pain modulation, butyrate, by promoting mucosal repair and reducing bowel inflammation, has been proposed to have an indirect effect on inflammatory visceral pain [23]. Moreover, SCFAs can stimulate free fatty acid receptors on epithelia cells, enterochromaffin cells (ECCs), immune, IPANS, vagus nerve and sympathetic nerve signaling. In this scenario, SFCAs, by regulating different immune, endocrine, epigenetic, neuronal and humoral mechanisms, may directly or indirectly impact the microbiota-gut-brain bidirectional communication axis. Preclinical studies have demonstrated that besides their peripheral actions, SCFAs have direct neuroactive properties in the CNS by crossing the BBB, although to a minimum extent [38]. Butyrate injection in rat and mouse brains enhanced brain-derived neurotrophic factor (BDNF) and glial-derived neurotrophic factor (GDNF) levels, favoring neurogenesis, synaptic plasticity, memory formation and mood-related behaviors, which are generally mediated by these factors [38]. SCFAs, by influencing tryptophan metabolism, may also modulate the synthesis of centrally active and mood-related neurotransmitters/neuromodulators, such as 5-HT and kynurenines [37]. Indeed, although in the human body the microbiota is not able to supply substantial amounts of tryptophan, which is mainly of dietary origin, the saprophytic microbial community is involved in the modulation of the amino acid metabolism and GF mice have higher plasmatic concentrations of tryptophan [37]. Furthermore, since the BBB is permeable to tryptophan but not to circulating 5-HT, the amino acid may represent a humoral source influencing serotoninergic neurotransmission in the CNS. An unbalanced kynurenine/tryptophan ratio and the shift of tryptophan metabolism from the 5-HT synthesis to kynurenine and kynurenine-derived metabolites are suggested to underlay development of anxiety and mood perturbations [39]. Interestingly, the enzymatic conversion of tryptophan into kynurenine may be promoted during gut inflammation by several factors such as cytokines and cortisol, but also by the perturbed microbiota [40,41]. During gut inflammation, enhanced plasmatic levels of kynurenine may favor its passage in the brain, where it is transformed into its metabolites, principally kynurenic acid and quinolinic acid [37]. Quinolinic acid is mainly produced by microglia and is described as a neurotoxic agent, which by acting as an agonist at glutamate NMDA receptors may participate in the development of depression, while kynurenic acid is an antagonist at NMDA receptors and is claimed to be neuroprotective [42]. Data from clinical investigations point to a role of the KynA/quinolinic acid ratio as an index of neuroprotection, and a reduced ratio is indicative of possible inflammation-induced depressive disorders [13].

The existence of a close connection between gut microbes and behavioral changes is supported by increasing evidence suggesting that the microbiota can produce host neurotransmitters (Figure 1). Recent studies support the concept that glutamate, the major excitatory neurotransmitter in the CNS, acts as a neuroactive molecule along the microbiota-gut-brain axis [36]. Although diet-derived glutamate represents the main source for the amino acid in the gut, some bacterial strains of the *Lactobacillus* spp. isolated, also present in some fermented foods, such as *Lactiplantibacillus*
*plantarum*, *Lacticaseibacillus casei* and *Lactococcus lactis* [43,44,45], synthesize the amino acid and express glutamate-sensing channels, suggesting that the amino acid may represent a signaling molecule participating in the host-microbiota crosstalk [46]. As a major excitatory neurotransmitter in the CNS, glutamate plays a fundamental role in the modulation of both physiological (e.g., memory, learning) and pathophysiological (e.g., stroke, epilepsy, neurodegenerative diseases, etc.) conditions [47]. Increasing evidence suggests that glutamate may also have a role in regulating gut motor and sensory functions via ionotropic and metabotropic receptors in the ENS [48,49,50]. Importantly, glutamate may behave as a sensory co-transmitter in IPANs, transmitting information from the mucosa to the ENS and in extrinsic primary afferents of sympathetic and vagal pathways projecting to higher centers, thus playing a role in the modulation of visceral sensitivity [49,50,51]. In a rat model of IBS induced by maternal separation, which is associated with alterations of the gut microbiota composition, the levels of glutamate AMPA receptors significantly increased in the hippocampus, and their blockade with the antagonist, CNQX, reduced visceral pain perception to colorectal distension, suggesting the possible involvement of ionotropic glutamate receptors in central mechanisms of chronic visceral pain control [52]. Ionotropic and metabotropic glutamate receptors are also involved in altered response to stress conditions. Stress-related changes in the microbiota-gut-brain axis may influence the expression and activity of glutamate receptors of the NMDA type as well as of BDNF, a neurotrophin involved in neuroplasticity, whose function is strictly correlated to NMDA receptor activation in different CNS regions [53,54]. In GF mice or antibiotic-treated dysbiotic mice, hippocampal BDNF and NMDA levels decreased, and were restored to control values by prebiotic galacto-oligosaccharide (GOS) treatment [33,55,56]. Moreover, a probiotic, such as *Bifidobacterium longum* NCC3001, upregulates BDNF, increasing neuronal plasticity in the ENS and reducing anxiety and depressive behavior in mice.

In both prokaryotes and eukarya, decarboxylation of glutamate by glutamate decarboxylase (GAD) yields gamma-amino-butyric acid (GABA), the most important inhibitory neurotransmitter in the brain. *Escherichia Coli*, *Pseudomonas aeruginosa* and specific strains that are generally recognized as safe or health-promoting, such as LAB (e.g., strains belonging to *Lactobacillus*, *Lactococcus* and *Streptococcus* genera) and *Bifidobacterium*, can synthesize GABA [57,58]. The functional relevance of the bacterial-produced GABA for the host-gut homeostasis is still unclear, although modulation of GABAergic pathways in the microbiota-gut-brain axis may influence the host health [14]. In rats, the oral administration of *Bifidobacterium*
*dentium*, which is able to synthesize GABA, inhibited visceral hypersensitivity, suggesting that targeting GABAergic signals along this microbiome-gut-brain axis may represent an approach for the treatment of abdominal pain [59]. In mice, oral administration of *Lacticaseibacillus rhamnosus* (JB-1) increased GABA levels in the CNS and was associated with reduced anxiety and depression-related behavior as well as with increased GABA receptor levels in the hippocampus; both the behavioral and molecular effects depended upon the integrity of vagal pathways, since both were abolished after vagotomy [59,60]. In mice, administration of a probiotic *E. coli* strain Nissle 1917 (EcN) was shown to produce an analgesic lipopeptide, C12AsnGABA OH, which facilitated GABA passage across the epithelial barrier, allowing the activation of GABA receptors on IPANs and inhibiting development of visceral hypersensitivity [61]. Besides the neuronal localization, GABA receptors localized in immune cells, such as dendritic cells, mast cells and T-cells, may modulate immune responses, including the downregulation of proinflammatory cytokine release [62,63]. 

In these latter views, studies aiming to clarify the functional dynamics and relevance of GABA and glutamate and their role as mediators of host-microbe crosstalk may eventually lead to the discovery of new molecules with potential therapeutic value for the treatment of visceral pain and microbiota-gut-brain-related CNS disorders.

Table 1 represents some of the effects of gut microorganisms on CNS, immune system and behavior.

Hormonal pathways are also involved in the bidirectional crosstalk in the microbiota-gut-brain axis (Figure 1). There are several reports suggesting that intestinal microbes may control the development and function of the HPA axis [55]. GF mice exhibit enhanced HPA axis activity, as well as elevated levels of plasma corticosterone, in response to stressful stimuli [18]. Corticosterone-induced postnatal stress in young adult BDNF heterozygous mice was associated with reduced BDNF levels and changes in NMDA receptor expression in the hippocampus [66]. Activation of the HPA axis as well as the release of ACTH and glucocorticoids is controlled by the vagus nerve as well as by the immune system, further supporting the existence of crosstalk between neuronal immune and hormonal pathways modulating the gut-brain communication [67,68]. *L. reuteri* ATTC-PTA 6475 upregulates plasma and brain concentration of oxytocin which restores social-interaction-induced plasticity in the ventral tegmental area neurons of autism spectrum disorders (ASD) mouse models [24]. Its cell surface components (peptidoglycan, teichoic acids, exopolysaccharides and other proteins) promote maturation of dendritic cells (DC), an immune modulation through the production of anti-inflammatory IL-10 [65].

Although more information is available on the ability of the enteric microbiota to influence host cell functions, there are indications that activation of descending efferent pathways from the CNS can modify the microbiota composition and function, for example via stress-mediator-induced gene expression or via the ANS. The ENS, by controlling intestinal motility, influences the intestinal microbiota composition by favoring the removal of exuberant bacteria from the lumen [69]. More recently, the ENS has also been suggested to influence the microbiota composition, since in zebrafish, genetic deletion of sox10-/-, which prevents ENS development, induced gut inflammation associated with development of “proinflammatory” microbial lineages [70].

## 4. Microbiota and Pain Regulation

The homeostasis between gut microbiota and host is important for the maintenance of a healthy gut barrier, preventing pathogen invasion for ENS and CNS development and for the regulation of the immune system. For these reasons, dysbiosis may contribute to a variety of conditions, including metabolic, cardiovascular and neurological diseases and GI disorders (i.e., irritable bowel syndrome—IBS, inflammatory bowel disease—IBD, celiac disease, food allergies). As reported above, there is an increasing body of evidence that microbiota is also important for modulation of pain through different peripheral and central mechanisms (Figure 1). 

### 4.1. Visceral Pain

Gut microbiota plays a key role in visceral and abdominal pain. A high proportion of patients with IBS show gut barrier dysfunction and an altered microbiota (i.e., reduced concentration of Bacteroides and Parabacteroides) compared to healthy volunteers [27,71]. Animal models pointed out that the use of antibiotics early in life produces long-lasting enhancement of visceral pain through alteration of gut microbiota, such as increased visceral hypersensitivity to colorectal distension, as observed in rats treated with vancomycin, whose effect was dependent on the time of exposure [5,72].

Luczynski et al. have demonstrated that visceral hypersensitivity in GF mice is associated with upregulation of TLRs and cytokines in the spinal cord, which are removed by postnatal colonization with microbiota. In GF mice, changes in the volume of brain regions involved in pain processing, such as the anterior cingulate cortex (ACC) and the periaqueductal gray (PAG), were observed, with a decrease of the ACC and an increase in the PAG volume [73]. It is noteworthy that colonization of microbiota restores the excitability changes of sensory neurons [5].

A number of clinical studies demonstrated that targeting gut microbiota may be a strategy for visceral pain management in GI disorders, stress response and altered behavior, including pain sensitivity [27,74]. In 2015, the non-absorbable antibiotic rifaximin received FDA approval for the treatment of diarrhea-predominant IBS [27]. Rifaximin, neomycin or specific probiotics (i.e., *L.*
*rhamnosus* GG or De Simone formulation which is a high-concentration probiotic preparation of eight live freeze-dried bacterial species [75]) may reduce stress-induced hyperalgesia, skeletal muscle hyperalgesia, neuropathic cutaneous mechanical allodynia and thermal hyperalgesia by altering the microbiota [76]. 

In addition, dietary strategies, such as low intake of foods high in fermentable oligosaccharides, disaccharides, monosaccharides and polyols (FODMAP) may have an impact on IBS symptoms in selected subjects by reducing fermentation, intestinal distention and dysbiosis [5]. 

A balanced microbiota may also have a beneficial effect on pain processing through peripheral and central actions. Indeed, patients with IBS have been shown to process visceral stimuli differently, with altered activation and deactivation patterns to noxious and non-noxious stimuli, which is typical of visceral hypersensitivity. Actually, these patients have deactivation of right insula, right amygdala and right striatum, altered activity of ACC and an altered descending modulation of visceral pain [27].

### 4.2. Inflammatory Pain

Inflammation decreases the pain threshold of nociceptors and increases the individual pain response. The inflammatory environment can lead to hyperalgesia (when noxious stimuli cause enhanced pain) and allodynia (when non-noxious stimuli cause pain). 

Furthermore, chemokines and cytokines released during the inflammatory process lead to activation of the intracellular downstream signal pathways (i.e., cyclic adenosine monophosphatase, protein kinase A (PKA), protein kinase C (PKC)) and subsequently to phosphorylation of receptors and ion channels in primary sensory neurons. This whole process results in neuronal hyperexcitability and peripheral sensitization [5]. 

Amaral et al. have demonstrated that inflammatory hypernociception induced by carrageenan, lipopolysaccharide, TNF-alpha, IL-1beta and chemokine CXCL1 is reduced in GF mice while hypernociception induced by prostaglandins and dopamine is similar in GF mice and conventional mice. It is also proved that carrageenan-induced inflammation is reduced in GF mice and it is reversed by restoration of microbiota or administration of LPS. Finally, an enhanced concentration of IL-10 is associated with decreased pain hypersensitivity in GF mice which can be reversed by anti-IL-10 antibody [77].

Another important study has been led by Guida et al., who also reported that vitamin D deficiency causes lower microbial diversity characterized by an increase in Firmicutes and a decrease in Verrucomicrobia and Bacteroidetes. It is noteworthy that vitamin D deficiency in mice induced mechanical allodynia associated with neuronal hyperexcitability [78]. 

Finally, it is important to highlight that the gut microbiota may exert important anti-inflammatory effects through the production of SCFAs (acetate, butyrate and propionate) and other microbial metabolites that restore normal gut permeability and increase T-reg cells and related cytokines. 

In rats, *L. rhamnosus* (LR-2) is able to reduce pain severity and cartilage destruction in induced osteoarthritis. These beneficial effects are related to the decreased expression of MCP-1 and IL-1β (two proinflammatory cytokines) and CCR2 (its receptor) and increased levels of IL-10 (an anti-inflammatory cytokine), GABA and PPAR-ʏ [64]. 

Based on these considerations, targeting gut microbiota by a dietary approach or probiotics supplementation may attenuate pain hypersensitivity in different inflammatory settings [5]. 

## 5. Peripheral and Central Mechanism of Pain Regulation

Inhibition of sensory-neuron-expressing ion channels and block of C- and Aβ-afferent fibers in the dorsal root ganglion (DRG) are considered a pivotal strategy to reduce hypersensitivity, hyperexcitability and persistence of pain. Mediators of gut microbiota (i.e., pathogen-associated molecular patterns (PAMPs), metabolites (SCFAs) and neurotransmitters (glutamate, GABA, 5-HT)) regulate the excitability of nociceptive DRG neurons acting on pain-related receptors or ion channels (i.e., TRLs, TRP channels, ionotropic and metabotropic glutamate receptors, GABA receptors) and reduce the activation of immune cells that release proinflammatory cytokines (TNF-α, IL-1β, IL-6) and chemokines (CCL2, CXCL1) [5]. SCFAs bind to FFAR2/3 (free fatty acid receptors), downregulating cytokines, eicosanoids and chemokines production by leucocytes [5]. 

Conversely, LPS binds to TLR4, leading to activation and sensitization of nociceptive neurons in DRGs. Furthermore, LPS can directly activate TRPA1 channel which releases CGRP and induces calcium flux and action potentials in nociceptive sensory neurons [5,79]. In addition, formyl peptides directly stimulate primary afferent nerves [80]. 

There is evidence to suggest that gut microbiota can also alter epigenetic processes such as histone acetylation, as SCFAs have histone deacetylase inhibitor activity. Actually, both early life and adulthood stress-induced visceral pain are related to epigenetic changes at the level of the spinal cord and at supraspinal level [81,82,83]. 

Besides, microbiota is able to produce neurotransmitters which can modify pain signaling. *Lactobacillus* spp., *B.*
*dentium* and *Bifidobacterium* spp. produce GABA through enzymatic decarboxylation of glutamate. GABA binds to GABA receptors expressed on the surface of DRG neurons leading to their depolarization which inhibits the nociceptive transmission [84]. Glutamate receptors, mainly of the NMDA type, participate in the transmission of visceral sensitivity from the small and large intestine, both locally, on extrinsic primary afferents, and in the CNS. In the rat, NMDA receptors were found in the soma of extrinsic primary afferent thoracolumbar DRGs, as well as in their peripheral terminals innervating the colonic mucosa [85]. Interestingly, NMDA receptors are largely co-expressed with capsaicin-sensitive TRPV-1, which is involved in visceral pain [85]. It is noteworthy that peripherally and centrally located NMDA receptors may contribute to development of visceral hypersensitivity in pathological conditions such as IBS.

*Candida* spp., *Streptococcus* spp., *Escherichia* spp. and *Enterococcus* spp. produce serotonin (5-HT) and the activation of 5-HT1 receptor causes a hyperpolarizing effect [86]. It is noteworthy that 5-HT has been implicated in different kinds of headaches, especially migraine [87]. In contrast, activation of 5-HT2 and 5-HT3 receptors has a depolarizing effect on primary nociceptive neurons [86]. 

Finally, microbiota is an important regulator of maturation, morphology and immunological function of microglia and central endothelial cells, pericytes, astrocytes and microglia, which are able to receive inputs from the GI tract. Hence, mediators released by microbiota may modulate neuroinflammation which is important for central sensitization to pain. TNF-α, IL-1β and CXCL-1 influence glutamatergic and/or GABAergic synaptic neurotransmission; these variations lead to central sensitization and consequent pain hypersensitivity [5].

## 6. Stress, Microbiota and Pain Amplification

Stress is an important modulating factor, which is able to change the GI environment through immune, neurochemical and physical mechanisms that convert the GI tract into a less favorable ecosystem for certain microorganisms. In this scenario, pain signals and visceral hypersensitivity increase [27]. 

Stress activates the HPA axis and the sympathetic nervous system and barrier dysfunction, causing increased permeability. In this condition, bacteria and bacterial antigens can cross the epithelial barrier, activate the mucosal immune response with the production of inflammatory cytokines and alter the composition of the microbiota [75]. 

In adult mice exposed to a social disruption stressor, an altered gut microbiota (i.e., reduction of *L.*
*reuteri* ATTC-PTA 6475) and increased levels of cytokines have been reported [88]. Finally, social stress promotes proinflammatory gene expression and monocyte differentiation, leading to an increased risk of inflammation-related diseases [89,90]. 

It is of note that early-life stress can influence the composition of microbiota, and disturbed bacterial colonization postnatally can alter pain pathways [27]. Furthermore, early-life stress produces a reduction of tight junction expression in the gut with increased gut permeability. The translocation of LPS, cytokines and bacteria causes the reduction of nociceptive threshold and the increase of neuronal excitability that contribute to hyperalgesia. Early life stress can also modify the nociceptor sensitivity by modulation of vagal afferent activity [76]. Interestingly, an increased risk of infantile colic has been reported in infants born preterm or who were submitted to antibiotic treatment in the first days of life [91].

## 7. Probiotics, Evidence of Efficacy in Pain Disorders

Probiotics are living microorganisms that give health benefits to the host when they are administered in an appropriate dose. A probiotic should also have antagonism against pathogens and provide optimal stimulation of the immune system [75]. However, to date, the beneficial effect of probiotics to prevent or treat diseases still needs to be demonstrated and fully clarified [75].

There is emerging evidence, mostly from preclinical studies, that the administration of probiotics attenuates stress-induced visceral hyperalgesia [92,93], neuropathic cutaneous mechanical allodynia and thermal hyperalgesia [94,95].

Microbiota produces proteases and protease inhibitors that can influence visceral perception [23]. Particularly, extracellular proteases (i.e., serine and cysteine proteases) activate protease-activated receptors (PARs). PAR-2 activation by mast cell tryptase and enterocyte-derived-trypsin-3 causes sustained hyperexcitability of DRG neurons which is implicated in visceral pain, while PAR-4 activation has an analgesic effect in vivo and in vitro by reducing DRG neuron excitability. Whether these protease effects are caused by direct action on mucosal cells or immune cells or directly on DRG nerve terminals is still uncertain [23]. As reported in a postinflammatory rat model for IBS, protease inhibitors (i.e., siropins) can reduce pronociceptive effects of proteases and visceral hypersensitivity [96].

Moreover, selected lactobacilli and bifidobacteria stimulate the production of secretory IgA and the activation of T regulatory cells and promote an anti-inflammatory response particularly important for gastrointestinal symptoms and tolerance in allergic diseases [97]. Different bacilli and Escherichia produce dopamine and norepinephrine which have reported antinociceptive effects on visceral sensitivity [23]. 

A recent study showed that the human and animal gut commensal bacterium *Clostridium butyricum* may provide improvement of visceral hypersensitivity of IBS through the inhibition of colonic inflammation in mice [98]. Zhang and co-workers found that *Roseburia hominis* reduces the abundance of butyrate-producing Lachnospiraceae, which is responsible for the development of visceral hypersensitivity in rats [99]. Benjak Horvat and colleagues reported that a selected probiotic combination of eight microorganisms (*Lactobacillus acidophilus*, *L. plantarum*, *L. casei*, *Lactobacillus delbrueckii* subsp. *bulgaricus*, *Bifidobacterium breve*, *Bifidobacterium*
*longum*, *Bifidobacterium*
*infantis* and *Streptococcus salivarius* subsp. *thermophilus*) decreased visceral hypersensitivity in rats, probably through the mast cell-PAR2-TRPV1 pathway [100]. The authors suggest that this probiotic combination may promote the host immune response, anti-inflammatory pathways and epithelial barrier function by modulating PAR2 expression in epithelial cells, GI smooth muscle cells and capsaicin-sensitive neurons and through its ability to regulate GI mucosa barrier functioning and inflammation [97,100,101]. In neonatal rats, *L. rhamnosus* GG has been shown to reduce chronic visceral pain after intracolonic administration of zymosan [102]. The probiotic also ameliorated visceral sensitivity and plasmatic concentration of corticosterone in rats stressed by maternal separation [92]. Different probiotics have been tested in patients with pain-related functional gastrointestinal disorders [103]. In a few randomized control studies [104,105], *L. rhamnosus* GG reduced the frequency and severity of abdominal pain in children with IBS and functional abdominal pain by reducing inflammatory cytokines and improving the gut barrier. In children with IBS, a multicenter, crossover RCT using a mixture of eight probiotic strains (*S.*
*thermophilus* BT01, *B. breve* BB02, *B. longum* BL03, *B. infantis* BI04, *L. acidophilus* BA05, *L. plantarum* BP06, *L. paracasei* BP07, *L. delbrueckii* subsp. *bulgaricus* BD08) reduced symptoms and improved quality of life significantly more than a placebo [106]. *Lactobacillus helveticus* R0052 and *B.*
*longum* R0175 regulated glucocorticoid negative feedback on the HPA axis and reduced stress-induced visceral pain [107]. *L. reuteri* ATTC-PTA 6475 has shown anti-inflammatory effects on human epithelial cells through the upregulation of an anti-inflammatory molecule, nerve growth factor (NGF), which increases the expression of TRPV1 in DRGs. [23,108]. *L. reuteri* 6475 has also been suggested to secrete histamine, which via H2 receptors may reduce intestinal inflammation, thus influencing development of visceral pain sensitivity. Indeed, the limited expression of H2R in patients with IBD may reduce the suppression of cytokine secretion by TLR [23]. In breastfed infants *L. reuteri* DSM 17938 significantly reduces crying time and colic, despite the cause of pain perhaps depending upon dysmotility, intestinal contractions or fermentation, lactose intolerance, food hypersensitivity, dysbiosis, stress or also a combination of the above [103,109].

*L. reuteri* DSM 17938 significantly reduced abdominal pain intensity and frequency of functional abdominal pain in six pediatric randomized control studies, as shown by a recent systematic review [110].

However, the American Gastroenterology Association does not recommend probiotics in children and adults with IBS because of significant heterogeneity in study design, outcomes, dosage and probiotics used [111].

Thanks to their immunomodulatory effects and anti-inflammatory benefits, probiotics have been recently considered to alleviate pain and improve quality of life, enhancing daily activities in patients with pain-related extraintestinal diseases such as rheumatoid arthritis [112]. 

## 8. Conclusions

Gut-brain axis is an essential bidirectional communication system between gut and brain which regulates homeostasis and many physiological functions of the GI tract, immune system and human behavior. Additionally, mediators of gut microbiota may also regulate neuronal excitability of the peripheral nervous system and nociceptors which are responsible for the onset of chronic pain. Moreover, microbiota plays a critical role in controlling depression, anxiety and stress response which are included in pain comorbidities. Dysregulation of the axis leads to a huge variety of diseases such as visceral hypersensitivity, stress-induced hyperalgesia, allodynia, inflammatory pain and functional disorders. 

Probiotics are a promising novelty which can reduce dysbiosis and contribute to the management of pain. The optimal dose, duration and probiotic strains still need to be identified.

## Figures and Tables

**Figure 1 cells-11-00971-f001:**
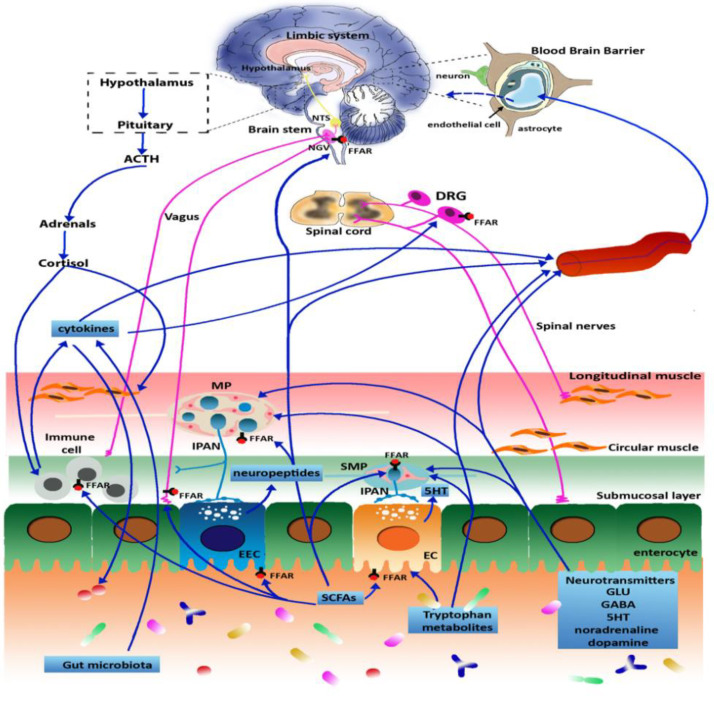
Schematic representation of the microbiota-gut-brain axis. The gut microbiota signals to the enteric nervous system (ENS) and to the CNS via different pathways, including endocrine, immune, metabolic and neuronal pathways as described throughout the text. In physiological conditions, the blood-brain barrier (BBB) allows the access of tryptophan metabolites and SCFAs into the CNS but blocks the passage of circulating neurotransmitters except for gamma aminobutyric acid (GABA). However, in pathological conditions, the disruption of BBB leads to an increased amount of circulating neurotransmitters in the brain.

**Table 1 cells-11-00971-t001:** Examples of gut microorganisms that are able to produce molecules with effects on CNS and behavior (↑ = increase; ↓ = decrease).

Gut Microorganism	Molecules, Metabolites and Neurotransmitters Involved	Effects on Gut-Brain Axis
*Bifidobacterium longum* NCC3001	↑ BDNF	↓ Anxiety and depressive behavior [30,58] ↑ Neuronal plasticity of ENS [58]
*Bifidobacterium dentium*	↑ GABA	↓ Visceral hypersensitivity [59]
*Lacticaseibacillus rhamnosus* JB-1	↑ GABA	↓ Anxiety and depressive behavior [59] ↓ Intestinal damage and inflammation [64]
*Escherichia coli* Nissle 1917	↑ C12AsnGABAOH	↓ Visceral hypersensitivity [61] ↑ Epithelial permeability of GABA [61]
*Limosilactobacillus reuteri* ATTC-PTA 6475	↑ Oxytocin	↓ Restores social deficits of ASD [24] Promotes DC maturation and immune modulation via IL-10 [65]

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
