# Peer review of "Microbiota and Pain: Save Your Gut Feeling"

_cells, 2022, doi:10.3390/cells11060971_

Round 1

Reviewer 1 Report

In my opinion paper is interesting and summarizing principal knowledge on pain and possible application of probiotics in its control.

I would like to recommend the paper for publication; however, extensive editorial corrections need to be performed by the authors.

Abstract: Maybe in the abstract authors can add a bit more, a few more lines related to the explored subject in the review.

Ln50: Please, add interval between reactions and [1,...

Ln81: "...TrKA receptors. Mediators..."

Ln82: Please, try to avoid use of slang, try to provide different explanation for "inflammatory soup" .

Ln156, 160: Please, add interval before [1]. Pleases, pay attention in other parts of the manuscript for similar typos.

Ln177: Maybe introduced abbreviation GIT for gastrointestinal tract?

Ln237: Please, use GI abbreviation

Ln286: [38]. Butyrate

Ln316: Italics for Lactobacillus and change to "spp.,"

Ln317: Please, add italics, for bacterial species, however, recommending to change name according to classification proposed in April 2020.

Lactiplantibacillus plantarum, Lacticaseibacillus casei

Ln345: Please, italics for bacterial species. Since this is a first-time mentioned Escherichia coli, name needs to be presented without abbreviation. In next occasion is only E. coli. What species of Pseudomonas are you talking about? If is Genus property, then need to be Pseudomonas spp.

Ln346-347: Italics for Lactobacillus, Lactococcus, Streptococcus, Bifidobacterium.

Ln350: Italics and fill name for B. dentium. IS any information in the cited reference work about strain number for this B. dentium culture? Please, provide it.

Ln353: Italics for L. rhamnosus, and since is the first time, provide full name according classification from 2020, If is available, provide strain number.

Ln358: Italics for E. coli. Do not need coma after strain and before Nissle. In fact, can be written only "E. coli Nissle 1917".

Ln368: Maybe will be better to change "summarizes " with "represent".

Table 1: Please add horizontal line on the top of the table. For GUT microorganism, remove italics from the strain numbers of B. longum NCC3001 and E. coli Nissle 1917, and add strain numbers for B. dentrum, L. rhamnosus, L. reuteri. Please, for Lactobacillus use new classification from 2020 (DOI: 10.1099/ijsem.0.004107). It is clear that arrow represent increase or decrease, but this needs to be specified in the legend of the table. Please, add a reference regarding all statements summarized in the Table 1.

Ln382: Italics for L. reuteri. And please, if is available, provide strain number from the cited study.

Ln425: Italics for the L. rhamnosus, change to new systematic name and ad strain number for this particular probiotic.

Ln455: Guida et al.,

Ln463: Italics for L. rhamnosus; if is available, add strain number for applied specific L. rhamnosus strain.

Ln490: Please, add italics for bacterial names; Since Bifidobacterium dentrium was already introduced earlier in this manuscript, in this place needs to be abbreviated.

Ln501: Italics for Candida spp., Steptococcus spp., Escherichia spp. and Enterococcus spp. Please, pay attention on spelling of Streptococcus

Ln523: Lactobacillus reuteri - abbreviated and check new taxonomy 2020. As well, provided a strain number of the particular reported Lactobacillus reuteri strain.

Ln554: Since in this context, you use English words (Lactobacilli and Bifidobacteria), they need to be written with non-capital L and B. 

Similar for Ln557 for Bacilli and Escherichia.

Ln559: Clostridium butyricum needs to be in italics.

Ln564-566: Italics for the bacterial species. L. delbrueckii subsp. bulgaricus; Streptococcus salivarius subsp. thermophilus. Please, check spelling of the bacterial names; ssp. needs to be change to subsp.

You have mentioned 8 microorganism, but are listed only 7. Please, correct this error.

Ln572, 576: Italics for L. rhamnosus.

Ln573: Add interval: ...zymusan [102].

In the section on Ln580-593: Pay attention for italics of bacterial names; Add "."after "B", "L". Ln582: L. Lactobacillus helveticus....> remove "L." .

Ln594: Italics for L. reuteri.

Please, check entire reference list if is in accordance with Instructions for Authors.

Author Response

Many thanks for your valuable comments. We completely agree with your suggestions, and we apologise for having left some inaccuracies in formatting while we submitted. We made the changes you requested and we implemented the abstract accordingly. 

Reviewer 2 Report

Morreale et al. have provided a very interesting and comprehensive review paper. It is prepared in a logical manner with a nice but somewhat poor layout. My minor comments are on the editorial stuff:
-Chapter names should not be at the bottom of the page, which is often the case in this manuscript
-Latin names of bacterial species should be written in italics 

Author Response

Many thanks for your suggestions. We have changed the editorial formatting of the manuscript as you recommended, both modifying the position of chapter titles and the writing of bacterial species.